# Unraveling the Impact of Microplastic–Tetracycline Composite Pollution on the Moon Jellyfish *Aurelia aurita*: Insights from Its Microbiome

**DOI:** 10.3390/microorganisms13040882

**Published:** 2025-04-11

**Authors:** Xuandong Wu, Hongze Liao, Xiaoyong Zhang, Zhenhua Ma, Zhilu Fu

**Affiliations:** 1Guangxi Key Laboratory for Polysaccharide Materials and Modifications, Guangxi Minzu University, Nanning 530008, China; xuandongwu@gxmzu.stu.edu.cn (X.W.); hzliao@gxmzu.edu.cn (H.L.); 2Guangxi Key Laboratory of Marine Natural Products and Combinatorial Biosynethesis Chemistry, Guuangxi Beibu Gulf Marine Research Center, Guangxi Academy of Sciences, Nanning 530007, China; 3University Joint Laboratory of Guangdong Province, Hong Kong and Macao Region on Marine Bioresource Conservation and Exploitation, College of Marine Sciences, South China Agricultural University, Guangzhou 510642, China; zhangxiaoyong@scau.edu.cn; 4Key Laboratory of Efffcient Utilization and Processing of Marine Fishery Resources of Hainan Province, Sanya Tropical Fisheries Research Institute, Sanya 572018, China; zhenhua.ma@scsfri.ac.cn

**Keywords:** *Aurelia aurita*, polyp, microplastics, tetracycline, microbiome

## Abstract

Microplastics have emerged as a pervasive marine contaminant, with extreme concentrations reported in deep-sea sediments (e.g., 1.9 million particles/m^2^) and localized accumulations near Antarctic research stations. Particular concern has been raised regarding their synergistic effects with co-occurring antibiotics, which may potentiate toxicity and facilitate antibiotic resistance gene dissemination through microbial colonization of plastic surfaces. To investigate these interactions, a 185-day controlled exposure experiment was conducted using *Aurelia aurita* polyps. Factorial combinations of microplastics (0, 0.1, 1 mg/L) and tetracycline (0, 0.5, 5 mg/L) were employed to simulate environmentally relevant pollution scenarios. Microbiome alterations were characterized using metagenomic approaches. Analysis revealed that while alpha and beta diversity measures remained unaffected at environmental concentrations, significant shifts occurred in the relative abundance of dominant bacterial taxa, including *Pseudomonadota*, *Actinomycetota*, and *Mycoplasmatota*. Metabolic pathway analysis demonstrated perturbations in key functional categories including cellular processes and environmental signal transduction. Furthermore, microplastic exposure was associated with modifications in polyp life-stage characteristics, suggesting potential implications for benthic–pelagic population dynamics. These findings provide evidence for the impacts of microplastic–antibiotic interactions on cnidarian holobionts, with ramifications for predicting jellyfish population responses in contaminated ecosystems.

## 1. Introduction

Microplastics (MPs), defined as plastic particles or fibers less than 5 mm in diameter, have emerged as a significant and pervasive environmental pollution issue [1]. They are globally disseminated through human activities, ocean currents, and runoff [2], with extreme concentrations found in deep-sea sediments (e.g., 1.9 million particles/m^2^) due to sedimentation processes and thermohaline-driven currents [3,4]. MPs are ecologically toxic to both terrestrial and marine ecosystems [5], with documented impacts including physical harm [6,7], oxidative stress [8], and chemical toxicity [9], and they cause disruptions to microbial communities and nutrient cycling [10]. In marine ecosystems, microplastics exert concentrated ecological impacts in specific regions (e.g., Antarctica and deep-sea habitats), where their effects are amplified by the heightened sensitivity of endemic species to pollution [4]. MPs accumulate not only on the ocean floor but also in pelagic food webs, with Antarctic krill (*Euphausia superba*) and benthic invertebrates showing high ingestion rates [4,11]. A study on benthic filter-feeders revealed that 42% of fanworm *Sabella spallanzanii* samples contained at least one MP particle [7], while Antarctic benthic communities exhibited even higher contamination, with 83% of macroinvertebrate samples containing MPs [11]. These particles can induce sublethal effects, including oxidative stress and metabolic disruption, and may fragment further into nanoplastics (<1 µm) within organisms, exacerbating bioaccumulation risks [4,12].

In addition to the inherent ecotoxicity of MPs, they can also serve as carriers for microorganisms, thereby altering the microbial composition in the environment to some extent [12,13]. Furthermore, MPs can transport composite pollutants, resulting in concentrated pollutant levels in localized areas. Numerous studies have indicated that MPs may facilitate the enrichment and dissemination of antibiotic resistance genes (ARGs) in the environment [14,15].

Researchers have reported that the intestinal microbiome can exhibit greater tolerance to tetracycline (TC) under the combined stress of MPs and TC [16]. In certain marine organisms, the combined effects of MPs and TC are more significant than exposure to either factor alone, leading to distinct effects on the microbiota. For instance, the combined effects of MPs and TC have a more profound effect on the marine medaka (*Oryzias melastigma*) than the individual effects of MPs or TC, with the intestinal microbial community being more affected than the gill microbial community [17]. A study on button corals has revealed that MPs can significantly enhance the diversity and richness of their symbiotic bacterial communities. Notably, when MPs are combined with higher concentrations of tetracycline, this promoting effect becomes even stronger. However, under low-concentration co-exposure conditions, both MPs and tetracycline synergistically reduce the abundance of dominant species, while the composition of dominant bacterial groups remains unchanged [18].

*Aurelia aurita* is a *Scyphozoan* with multiple life stages, each occupying distinct ecological positions within the ecosystem [19]. For example, the polyp stage is benthic, whereas the adult (medusa) stage is planktonic [20]. The microbiome within the body is crucial for life activities, particularly for facilitating life history transformations [21,22]. The composition of dominant bacterial species in *Aurelia* varies across different life stages, with shifts in the most abundant and highly ranked microbial populations [23]. Without a healthy microbiome, *A. aurita* would not be able to undergo transitions from polyp to strobila [21,22]. Studies combining transcriptomic and microbiological analyses have demonstrated that stimulation by different microorganisms can regulate the *A. aurita* transcriptome by influencing related gene fragments, including critical biological processes like morphogenesis, development, the stress response, and those involved in quorum quenching [24]. Notably, such dysbiosis further induces apoptosis and immune activation, unequivocally highlighting the microbiota’s essential protective role in maintaining host homeostasis. Crucially, such dysbiosis triggers a rewiring of specific metabolic pathways, a molecular-level restructuring that often escapes detection through conventional phenotypic observation but becomes strikingly evident via transcriptomic profiling.

The composition of the microbiome community in *Aurelia* varies by region and subspecies, typically comprising between two and five dominant taxa [25]. Previous studies have identified *Rickettsiales* and *Mollicute* as the two major components of the microbiome in *A. coerulea*, with the genomes of *Rickettsiales* and *Mycoplasma* significantly reduced after numerous generations of symbiosis, leading them to rely heavily on *A. coerulea* for metabolic energy and vitamins. Despite geographical barriers, *Mollicute* has a relatively high affinity for *Aurelia* populations worldwide [23]. Additionally, the ratio of *Mollicute* to *Rickettsiales* differs across various life stages. Microbial analyses of the surface and interior of *A. solida* polyp, conducted using scanning electron microscopy (SEM) and transmission electron microscopy (TEM), detected only bacteria of the genus *Mycoplasma* [26]. Furthermore, fluorescence in situ hybridization (FISH) analysis of *A. aurita* revealed that the bacteria were distributed on the mucus and epithelial surfaces of the polyp, suggesting an endosymbiotic relationship between *Mycoplasma* and *A. aurita* [25].

Currently, there are no reports on structural changes in the bacterial flora in the polyps of *Aurelia* induced by MPs without pollutants. Some studies have examined the effects of other environmental factors such as salinity [27], radiation [28], and pH [29] on the symbiotic microbial communities of *Aurelia*, revealing certain changes in bacterial species. However, these studies have indicated that changes in community diversity are not significant under certain stimuli. This provides a foundational understanding of the effects of stress on the *Aurelia* microbial community.

Although numerous studies have explored the relationship between MPs and the microbiome [15,17,18,30], to date, there has been no research investigating the impact of MPs on polyps of the moon jellyfish using microbiome analysis. Based on previous research, our experiment selected *A. aurita* polyps to utilize bioinformatics tools to assess whether long-term exposure to MPs and tetracycline could affect the diversity of the coexisting microbiome and influence metabolic pathways, ultimately revealing differences in the responses of *A. aurita*. This study aimed to provide data and theoretical support to evaluate the effects of MPs and MPs/TC complexes on jellyfish.

## 2. Materials and Methods

### 2.1. Stock Culture of Polyps

Polyps of *A. aurita* were obtained from the Hongyuan breeding base in Beihai, and the parent medusae were collected from the coast of Hong Kong. Through joint phylogenetic analysis of mt-COI and 16S gene fragments, the specimens were confirmed as *A. aurita* [31].

Before the experiment, polyps were cultured in flat plastic cylindrical containers with a diameter of 134 mm and depth of 83.4 mm, containing 500 mL of artificial seawater at a salinity of 30 psu, (pH: 8.8, DO: 5 mg/L) and maintained at 25 °C in the dark. Artificial seawater was prepared by mixing sea salt, using Tropic Marin^®^ Pro-reef seasalt (Tropic Marin^®^, Fulda, Germany), with ultrapure water. The polyps were fed ad libitum with newly hatched *Artemia* sp. nauplii (Stock Culture of Polyps, JiangYuRen biotechnology company, Beijing, China) for 3–4 h, once or twice weekly, after which the artificial seawater was replaced.

### 2.2. Experimental Design

A two-factor orthogonal experiment was conducted to investigate the effects of prolonged exposure to varying concentrations of MPs and tetracycline hydrochloride (TCHCl) on *A. aurita* polyps. The concentrations of MPs were set at three levels (0, 0.1, and 1 mg/L), and tetracycline hydrochloride concentrations were set at three levels (0, 0.5, and 5 mg/L), resulting in a total of nine treatments (Table 1), with each group consisting of approximately 100 individuals and three replicates. The MPs used were spherical polystyrene (PS) particles, 80 nm in size (Tianjin Bessler Chromatography Technology Development Center, Tianjin, China). The monodispersity and morphological characteristics of the 80 nm PS microspheres were verified by transmission electron microscopy (TEM). Tetracycline hydrochloride was obtained from Shanghai McLean Biochemical Technology Co., Ltd., Shanghai, China. During the experiment, the polyps were cultured in a light-proof incubator, and the temperature was maintained at 23 ± 1 °C. The polyps were fed with newly hatched *Artemia* sp. nauplii (ca. 300 ind.^−1^/L) once a week, and the culture water was changed 1–2 h after feeding. The experiment lasted for 185 d, after which the polyps were collected using a dissecting needle, rinsed twice in PBS and again in ultrapure water before being quickly frozen in liquid nitrogen. The experiment lasted for 185 days. At the endpoint, polyps were carefully collected using a sterile dissecting needle by detaching them from their basal attachments. The isolated polyps were then gently rinsed twice in PBS (4 °C), followed by a final wash in ultrapure water to minimize contamination (e.g., microplastic adhesion) that could affect downstream analyses. Finally, the rinsed polyps were transferred to cryovials and flash-frozen in liquid nitrogen for storage.

### 2.3. 16S rRNA Amplicon Sequencing and Analysis

Total DNA was extracted using the CTAB method, and primers specific for the bacterial 16S rDNA V3V4 region, 338F (5′-ACTCCTACGGGAGGCAGCAG-3′) and 806R (5′-GGACTACHVGGGTWTCTAAT-3′) [32], were utilized. The PCR protocol was performed as described by Cui et al. [32]. PCR product purification was performed using magnetic beads (Vazyme VAHTSTM DNA Clean Beads, Vazyme Biotech, Nanjing, China). The PCR amplification products were quantified via fluorescence using the Quant-iT PicoGreen dsDNA Assay Kit (Thermo Fisher Scientific, Waltham, MA, USA), with measurements taken using a Microplate reader (BioTek, FLx800, BioTek Instruments, Winooski, VT, USA). Based on the fluorescence quantification results and the required sequencing amount for each sample, samples were mixed in appropriate proportions.

### 2.4. Sequencing Library Construction

A sequencing library was prepared using an Illumina TruSeq Nano DNA LT Library Prep Kit (Illumina, San Diego, CA, USA). BECKMAN AMPure XP Beads (Beckman Coulter, Brea, CA, USA) were used to remove adapter self-ligating fragments through magnetic bead screening and purification of the library after adapter addition. The DNA fragments connected to the adapter were PCR-amplified to enrich the sequencing library templates, and the enrichment products were further purified using BECKMAN AMPure XP Beads. Finally, the library was selected and purified by 2% agarose gel electrophoresis.

Before sequencing, the library was subjected to quality checks on an Agilent Bioanalyzer using an Agilent High-Sensitivity DNA Kit (Agilent Technologies, Santa Clara, CA, USA). Subsequently, the library was quantified using the Promega QuantiFluor fluorescence quantitative system and the Quant-iT PicoGreen dsDNA Assay Kit (Thermo Fisher Scientific, Waltham, MA, USA). Double-end sequencing was performed using a MiSeq sequencer with a MiSeq Reagent Kit V3 (Illumina, San Diego, CA, USA) (600 cycles).

### 2.5. Data Analysis

Bioinformatics analysis was conducted using QIIME2 (2019.4), R language, ggtree, and phyloseq with the Greengenes database (Release 13.8, https://greengenes.lbl.gov/, accessed on 1 May 2024) as a reference [33]. The classify-sklearn algorithm (https://github.com/QIIME2/q2-feature-classifier, accessed on 1 May 2024) was used for analysis [34]. For the feature sequences of each amplicon sequence variant (ASV), the default parameters in QIIME2 were employed, and species annotation was performed using a pre-trained Naive Bayes classifier.

## 3. Results

### 3.1. Microbiome Composition

The phylogenetic tree plot (Figure 1) revealed that the two phyla with the highest abundance in the 27 polyp samples were *Mycoplasmatota* and *Pseudomonadota*, with the three largest classes within *Pseudomonadota* being *Alphaproteobacteria*, *Gammaproteobacteria*, and *Deltaproteobacteria*, in descending order. Notably, *Alphaproteobacteria* comprised approximately 84% of the total *Pseudomonadota*, whereas *Mycoplasmatota* consisted entirely of the genus *Mycoplasma*.

In the treatment groups exposed only to MPs (ASW, LMPs, and HMPs) (Figure 2a), *Mollicutes* in the polyp samples of *A. aurita* exhibited an initial increase followed by a decrease. The values of the HMP group were significantly lower than those of the LMP group (*p* < 0.05). *Alphaproteobacteria* demonstrated a pattern of first declining under low concentration of MPs and then rising at higher level of MPs. Both *Acidimicrobiia* and *Thermolephilla* decreased as the concentration of microplastics increased. *Chloroflexi* demonstrated an increasing relative abundance at higher MP concentrations. *Actinomycetota* exhibited a unimodal response to microplastics, with its abundance initially increasing under low microplastic concentrations (LMPs) compared to the ASW control, followed by a subsequent decrease at high microplastic concentrations (HMPs).

A comparison of the low- and high-concentration tetracycline groups (Figure 2b) showed that high concentrations of tetracycline significantly reduced the relative abundance of *Mycoplasma* and *Rickettsiale*, and this trend continued after the addition of compound MPs (from LTLP to HTLP and from LTHP to HTHP). In the HTLP group, *Rickettsiale* constituted approximately 4%, whereas in the HTHP group, it accounted for approximately 21%.

After compounding low concentrations of tetracycline in different MP concentration groups (i.e., LTC, LTLP, and LTHP) (Figure A1), the relative abundance of *Gammaproteobacteria*, *Actinomycetota*, and *Chloracidobacteria* decreased as the MP concentration increased. In contrast, the relative abundance of *Thermolephilla* initially increased and then decreased, differing from the continuous decline observed in the MP-only treatment group. Concurrently, *Mycoplasmatota* exhibited a trend of first decreasing and then increasing when combined with low-concentration tetracycline, whereas *Alphaproteobacteria* increased with higher MP concentration, which also contrasted with the behavior in the MP-only treatment group.

### 3.2. Alpha Diversity Analysis

In the alpha diversity index (Figure 3), coverage was relatively high, exceeding 99.6% in each group. The MP treatment groups (LMPs and HMPs) reduced the diversity of the microbial flora in *A. aurita* polyps (*p* > 0.05), whereas tetracycline increased the diversity of bacterial flora in the polyps (*p* > 0.05); however, these changes were not statistically significant.

### 3.3. Beta Diversity Analysis

The β-diversity analysis diagram (Figure 4) indicated that non-metric multidimensional scaling (NMDS) analysis revealed a significant overlap between the ASW and low-tetracycline-concentration treatment groups (*p* > 0.05). The other treatment groups combined with pollutants were distinctly different from ASW. The low-concentration-MP treatment also differed from the other treatments in its coordinate axis position (*p* > 0.05). Furthermore, after combining different concentrations of MPs with tetracycline, there were overlaps in community diversity, suggesting that even with varying concentrations of MPs and tetracycline, community composition remained partially consistent.

### 3.4. Species Difference Analysis and Landmark Species

A Venn diagram (Figure 5) shows that after the introduction of low concentrations of MPs (i.e., the LMP group), the number of symbiotic ASV/OTUs in *A. aurita* polyps significantly decreased. However, in the HMP group, the number of ASV/OTUs increased again.

The histogram of linear discriminant analysis (LDA) effect values of the marker species (Figure 6) indicates that the LDA score for *Xanthomonadaceae* was higher in the MP treatment group, whereas the LDA scores for *Vibrionaceae* and Roseovarius were higher in the ASW group.

The association diagram (Figure 7) showed a strong correlation between *Mycoplasma* and both *Ruegeria* and unidentified *Rickettsiales*, whereas *Vibrio* exhibited a high correlation with *Nitrospiraceae*.

### 3.5. Species Indicator Analysis

Species indicator analysis (SIA) reveals the environmental adaptability of the symbiotic microbiota in *A. aurita* polyps by identifying microbial taxa that are significantly enriched under specific pollution conditions. Experiments demonstrate that the polyp-associated microbial communities exhibit distinct variations under exposure to microplastics (MPs), tetracycline (TC), and combined pollution (MPs + TC). Key indicator species for different pollution types include *Rickettsiales*, *Pseudomonas stutzeri*, and *Photobacterium*, among others (Figure 8). The SIA results confirm that structural shifts in the polyp microbiota can directly reflect the type and intensity of environmental stressors.

In the high/low microplastic concentration groups (HMPs/LMPs), SIA detected a significant enrichment of *Rickettsiales*, *Chromatiales*, and *Acidimicrobiales*.

Among the top 30 indicator species, the HTHP group (high MPs + high TC) showed the highest number of significant indicator taxa (Figure 9). Within the HTHP group (high MPs + high TC), *Photobacterium*, *Planctomycetes*, and *Rhizobiales* emerged as the most robust indicator taxa.

### 3.6. Pathway Differences

A joint analysis of KEGG and MetaCyc databases (Figure A2) indicated that the combined effects of MPs and tetracycline on *A. aurita* polyps primarily affected their metabolism, degradation, biosynthesis, and resistance to a limited number of antibiotics. Within metabolism and degradation, amino acid metabolism exhibited the most significant change in relative abundance, whereas the metabolism of exogenous substances was the most frequently affected. Replication and repair were the processes most affected in terms of genetic information processing. Additionally, KEGG pathway analysis suggested that MPs and tetracycline influenced the organic system, including environmental adaptation and development.

A comparison of KEGG pathways (Figure A3) revealed that MPs exerted varying effects on the pathways of symbiotic microorganisms in *A. aurita* polyps, including photosynthesis, the Wnt signaling pathway, and the biosynthesis of cell membrane signaling molecules. In contrast, tetracycline primarily affected ethylbenzene degradation. When both were combined, changes in MP concentration led to alterations in biosynthesis and metabolism-related pathways.

## 4. Discussion

### 4.1. Changes in Microbiome

The polyps of *A. aurita* hosted a significant number of cell-associated bacteria, primarily *Alphaproteobacteria* and *Mycoplasma*, which together comprised approximately 80% of the total bacterial population, consistent with previous study [23]. MPs without tetracycline can alter the relative abundance of cell-associated bacteria. The metabolic activity of these symbiotic bacteria is closely linked to the physiological state of their *Aurelia* host, suggesting that shifts in microbial composition may reflect underlying stress responses in *A. aurita* under pollutant exposure. Notably, in the single microplastic (MP) treatment groups (ASW, LMPs, and HMPs), the relative abundance of *Mycoplasma* increased in the LMP group (Figure 2), while the *Mollicute*/*Rickettsiales* ratio was higher in the LMP group than in the HMP group. This implies that low MP concentrations may favor *Mollicutes*, possibly due to differential tolerance or resource competition among bacterial taxa. When tetracycline was introduced (LTLP to LTHP and HTLP to HTHP), the *Mollicute*/*Rickettsiales* ratio remained elevated at low MP levels, suggesting that antibiotic exposure further amplifies this microbial imbalance under mild MP stress. Cladogram analysis (Figure A4) confirmed that MP concentrations significantly influenced the relative abundance of key bacterial strains across samples, though these effects were dynamic rather than consistent, highlighting the complexity of microbiome responses to combined pollutant exposures. However, this effect was not constant. Certain bacteria exhibited a low relative abundance in environments with low MPs, whereas their abundance increased in high-MP environments. This suggests that the regulation of endophytic flora by MPs could be dual, displaying different behaviors at varying concentrations.

Previous gut microbiome research has indicated that a decrease in *Alphaproteobacteria* is associated with increased concentrations of MPs [30]. However, this experiment demonstrated a different trend, likely due to the selection of endogenous symbiotic microorganisms rather than focusing only on material exchange from intestinal microorganisms or mucus. The *Alphaproteobacteria* detected were predominantly plasmid-free, cell-associated types, resulting in a higher ratio of *Alphaproteobacteria* to *Mycoplasma* compared to the data reported in some studies on polyps of the genus *Aurelia*.

After the addition of MPs and tetracycline, the diversity of the endomicrobial community in *A. aurita* polyps showed no significant changes in either alpha or beta diversity, whereas the relative abundance of the dominant bacterial community varied to some extent. This may be attributed to the dominant bacterial species comprising a large proportion of the total bacterial community, whereas other species represented a smaller fraction. Previous studies on the microbial community of *Aurelia* polyps showed no significant change in alpha diversity under ocean acidification [29], but under radiation exposure, the community composition shifted significantly [28]. These findings suggest that this level of stimulation cannot induce substantial fluctuations in the population structure, thereby confirming the stability of the community. Certain bacterial species, such as *Stenotrophomonas* and *Pseudomonadaceae*, can exhibit high relative abundances only in low-MP environments, indicating that their tolerance to MPs is limited and that their abundance decreases under higher-concentration stimulation. The analysis of the LTC and HTC groups demonstrated that the strains significantly affecting the relative abundance were primarily identified in the treatment group with high concentrations of tetracycline hydrochloride, suggesting that low tetracycline concentrations exerted a lesser effect on the relative abundance of endophytic bacteria in *A. aurita* polyps. Furthermore, low tetracycline concentrations were less likely to significantly affect epiphytic microorganisms in the moon jellyfish. Notably, *Phaeobacter gallaeciensis* maintained a relatively high abundance in both the HTC and HTHP groups, suggesting its potential resistance to MP environmental pollution and capacity to compete with other pollutants in high MP environments. An accumulation of *Phaeobacter gallaeciensis* has also been observed in MPs along the coast of Shenzhen [35].

When there are multiple pollutants, the effect on cell-associated bacteria is different from the pattern shown by a single pollutant. For example, when there is microplastic pollution only, there was no significant difference in the abundance of *Mycoplasma* in the ASW and LMP groups. However, after being exposed to high concentrations of tetracycline at the same time, that is, comparing the HTC and HTLP groups, the values found for HTLP were significantly higher than those obtained for HTC. Similarly, when comparing the three groups of low microplastic concentrations of LMPs, LTLP, and HTLP, the abundance of the *Mycoplasma* genus in the LMP group was also higher than that in the other two groups. This is different from the results of the tetracycline treatment alone. In the tetracycline treatment alone, only HTC was significantly lower than ASW and LTC, and there was no significant difference between ASW and LTC.

### 4.2. Alterations in Metabolic Pathways

The differences in bacterial species and pathways observed with single pollutants compared to combined pollutants indicated that the interactions among the combined pollutants were more complex than a simple additive relationship, leading to significant changes in the bacterial community within the polyps of *A. aurita*. This complexity ultimately affected the metabolic pathways and functions of *A. aurita*, including the metabolism of exogenous substances. Utilizing the KEGG and MetaCyc databases provided a comprehensive understanding of how microplastic composite pollutants affected the microbiome of *A. aurita* polyps, revealing that these composite pollutants significantly influenced the metabolism and synthesis pathways. The stimulation of MP composite microorganisms activated the degradation pathways for foreign substances within the *A. aurita* microbiome, thereby affecting the metabolism of energy substrates. Under laboratory conditions, *A. aurita* can be fed *Artemia* sp. nauplii, which is high in protein. In natural environments, *Aurelia* polyps can primarily consume plankton, including zooplankton such as copepods [36] and small proportions of phytoplankton such as algae [37]. Different types of food sources can significantly affect the altered pathways and may lead to variations in their reproduction rates in polyps of the genus *Aurelia*. For instance, polyps that consume ciliates tend to exhibit higher bud production rates than those fed on *Artemia* [38]. Generally, phytoplankton have a lower protein content but a higher lipid content than zooplankton. Consequently, after ingesting MPs in natural environments, not only protein metabolism but also additional metabolic pathways, such as those related to lipids, may be affected. Our previous metabolomics study [39] on the effects of microplastics and tetracycline on *A. aurita* polyps revealed that these two pollutants induced highly similar alterations in metabolic pathways related to amino acids, nucleotides, carbohydrates, and lipids—findings that are strongly consistent with the current study. Notably, some of the signaling pathways affected also showed this same pattern of similarity.

### 4.3. Potential Life Stage-Changing Factor

Endophytic bacteria play a crucial role in the life cycle transition of *A. aurita* from asexual to sexual reproduction. The comparison between HMP and LMP concentrations can affect the Wnt pathway, which is vital for regulating the transformation of polyps into the strobila life stage [40]. By modulating the Wnt pathway, the strobilation rate in the genus *Aurelia* can be determined, influencing whether strobilation occurs at all. Additionally, the ratio of *Mollicute*/*Rickettsiales* initially increased and then decreased with increasing microplastic concentrations. In the ephyrae life stage of *Aurelia*, the ratio of *Mollicute*/*Rickettsiales* is slightly lower than that in the polyp stage, whereas it reaches its highest value in the medusa stage [23]. Both observations suggest that different concentrations of MPs may alter endophytic microorganisms in *A. aurita*, leading to varying sensitivities in their life stage transitions. In marine environments, MPs may respond differently to Wnt stimulation when combined with various pollutants, potentially altering the microbial ratios within their bodies and affecting the reproduction of the genus *Aurelia*. This is particularly relevant when the environment contains pollutants that act as regulators of life stage transitions, such as those that disrupt retinoid physiology. During the transition from polyps to strobila, retinol, derived from both endogenous synthesis and exogenous sources, promotes this transformation [41]. Evidence has indicated that various environmental pollutants, including pesticides and polychlorinated biphenyls (PCBs), can affect biological retinol regulation [42]. Plasticizers added to MPs, such as bisphenol A, can modulate the mRNA levels of retinol-related receptors in murine embryos [43]. In addition to the individual effects of pollutants, researchers have identified the combined effects of MPs and pollutants on marine organisms. For instance, a study on *Peneaus vannamei* demonstrated that the combination of MPs with di-(2-ethylhexyl) phthalate could significantly affect retinol metabolism. When combined with metabolomics, it can also be seen that the combination of microplastics and tetracycline affects indole and its derivatives and retinol in the metabolites of *A. aurita* polyps. Indole and its derivatives and retinol will have a certain impact on the life stage transition of the polyp of moon jellyfish [39].

### 4.4. Ecological Impact

The environmental concentrations of microplastics and tetracycline used in the experiment were close to those found in seawater, particularly in sediments [44,45]. The concentration of microplastics in natural water bodies generally falls within the range of 0 to 0.1 mg/L [46]. In contrast, sediment pore water can exhibit significantly higher concentrations, with estimates suggesting levels as high as 162 mg/L [47]. In some sediments, microplastics can account for up to 3.3% of the total weight [48]. Regarding tetracycline, its concentration in seawater typically ranges from 5 ng/L to 2500 ng/L, with peak levels reaching up to 12 mg/L near sewage treatment plants [49]. Tetracycline also tends to accumulate in sediments, with adsorption increasing in the presence of higher organic carbon content [50]. Two gradient concentrations were used for both microplastics and tetracycline in order to better simulate specific ecological effects in the environment.

A previous study [51] has shown that *A. aurita* may also serve as a potential vector for the transmission of pathogenic microorganisms like *Vibrio*. Compared to the control group, *A. aurita* exhibited higher antimicrobial resistance (AMR) to multiple antibiotics, including tetracycline, thereby posing a greater ecological risk when blooming occurred. In marine ecosystems, *Vibrio* plays multiple roles: as a component of the ecosystem, it participates in the degradation and cycling of organic matter [52,53]. It serves as a symbiotic bacterium for certain marine organisms (e.g., shellfish and corals), aiding in digestion or providing protection [52]. It can colonize host organisms and may form biofilms [54]. However, it is also an important pathogen in aquaculture, causing diseases in various commercially farmed aquatic species [55]. Our data showed *Vibrio* is one of the dominant genera in the polyp stage of *A. aurita*. However, whether specific strains are harmful to other marine organisms requires further investigation. In this study, compared to the ASW group, tetracycline treatment did not significantly reduce the abundance of *Vibrio*. However, higher concentrations of tetracycline significantly increased the abundance of *Vibrio* compared to lower concentrations when comparing the data for the HTC to LTC groups. This may be due to the fact that high concentrations and prolonged exposure are more likely to trigger the spread and accumulation of tetracycline resistance genes. In natural environments (especially sediments), *Vibrio* strains can acquire high antibiotic resistance through multiple mechanisms, and the distribution of their resistance profiles is closely associated with habitats [56].

Based on previous studies on the accumulation of tetracycline resistance fragments in microplastics [14,15], we hypothesized that exposure to microplastics would increase the abundance of *Vibrio* in *A. aurita*. However, the experimental results showed the opposite trend. Under microplastic treatment alone, the concentration of microplastics had no significant effect on the abundance of *Vibrio*, and no significant changes were observed compared to the control group. This may be because the microplastic concentrations used in the experiment were insufficient to significantly affect the abundance of *Vibrio*.

Under high tetracycline conditions, the mean abundances of *Vibrio* in the HTLP and HTHP groups, which were treated with microplastics, were similar, indicating comparable *Vibrio* abundances under these two conditions.

In the polyp stage of *A. aurita*, *Vibrio* also showed a strong association with an unidentified *Nitrospiraceae*. This suggests that the nitrate provided by *Nitrospiraceae* may support the growth of *Vibrio*, thereby increasing its abundance in the polyp body.

The observed shifts in dominant bacterial taxa suggest potential microbial interactions, though current technical limitations—such as unculturable strains and the complexity of quorum-sensing mechanisms—preclude definitive mechanistic insights. The ecological roles of these uncultured microorganisms remain critical to address. Future studies integrating simplified in vitro co-culture with metagenomic binning could unravel metabolic cross-feeding dynamics, pollutant-driven quorum-sensing network remodeling, and microbe–host co-adaptation, advancing our understanding of holobiont–environment interactions in cnidarians.

## 5. Conclusions

This study reveals that microplastics and tetracycline at environmental concentrations selectively modify the symbiotic microbiome of *Aurelia aurita* polyps, particularly affecting dominant taxa like *Pseudomonadota* and *Mycoplasmatota*, while maintaining overall microbial diversity. These shifts in bacterial composition correlate with altered metabolic pathways and potential impacts on life stage transitions. The persistence of *Vibrio* under antibiotic exposure raises concerns about resistance propagation. These findings underscore the subtle yet ecologically significant effects of anthropogenic pollutants on cnidarian–microbe symbioses, highlighting the need to consider microbiome-mediated consequences for marine ecosystems facing increasing pollution pressures.

## Figures and Tables

**Figure 1 microorganisms-13-00882-f001:**
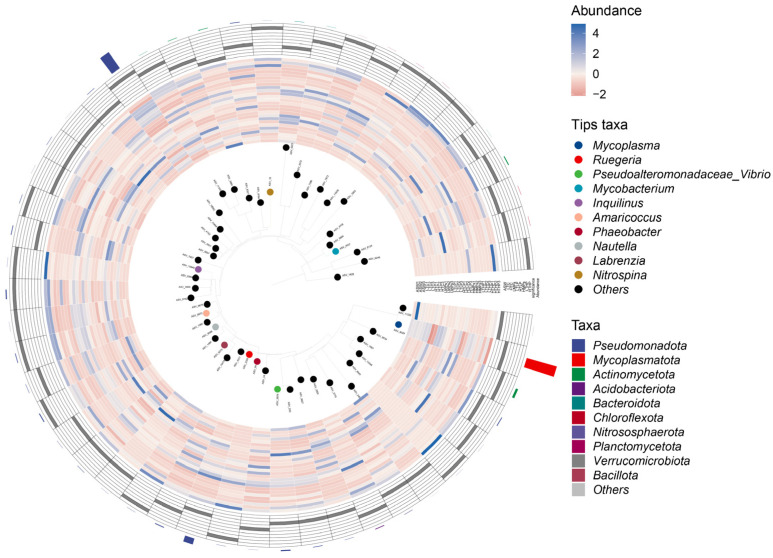
Phylogenetic tree of all collected samples. Evolutionary relationships are presented in a donut diagram format. The key elements of the diagram include (1) the phylogenetic tree, which defaults to ASV characteristic sequences or OTU representative sequences (tips of the diagram) and their connected branches, colored by genus classification; (2) an abundance heat map by sample; (3) a differential abundance heat map, highlighting the group with the highest abundance for each tip species, with significance tested using the Wilcoxon rank sum test (*p*-value < 0.05 after FDR correction using the “BH” method); and (4) a bar chart showing the total abundance of each tip, colored by phylum.

**Figure 2 microorganisms-13-00882-f002:**
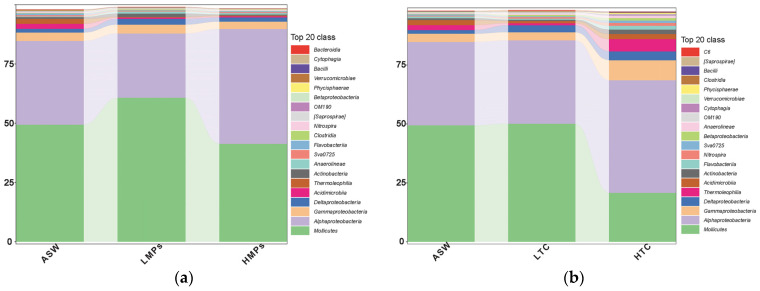
(**a**) Single MP treatment group (ASW, LMPs, HMPs); (**b**) single TCHCl treatment group (ASW, LTC, HTC).

**Figure 3 microorganisms-13-00882-f003:**
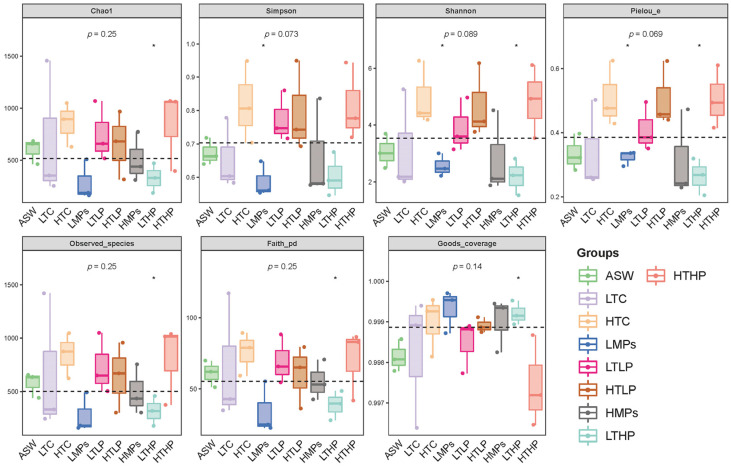
Alpha diversity index grouped box plot. Each panel represents a different alpha diversity index, indicated in the gray area at the top. In the box plot, the elements are as follows: the upper and lower edges of the box represent the interquartile range (IQR), the line inside the box is the median, and the whiskers extend to 1.5 times the IQR (indicating the maximum and minimum inner values). Points beyond the whiskers are outliers. Numbers below the diversity index labels indicate *p* values from the Kruskal–Wallis test. * represents for *p* < 0.05.

**Figure 4 microorganisms-13-00882-f004:**
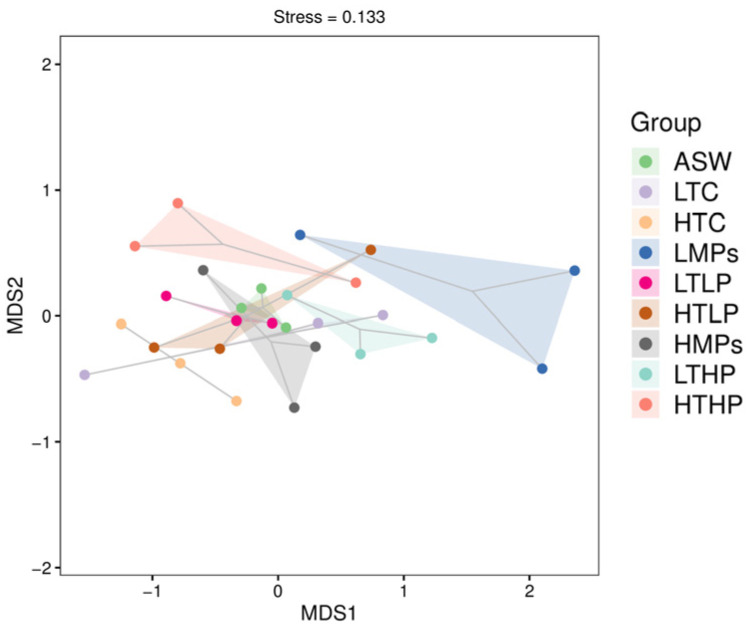
Two-dimensional non-metric multidimensional scaling (NMDS) plot of all samples. Each point represents a sample, with different colors indicating different groups. To maintain visual balance, the physical unit lengths of both coordinate axes were set to a ratio of 1 by default. NMDS, which is based on rank sorting, illustrates that smaller distances between points correspond to more similar microbial communities, whereas larger distances indicate greater differences.

**Figure 5 microorganisms-13-00882-f005:**
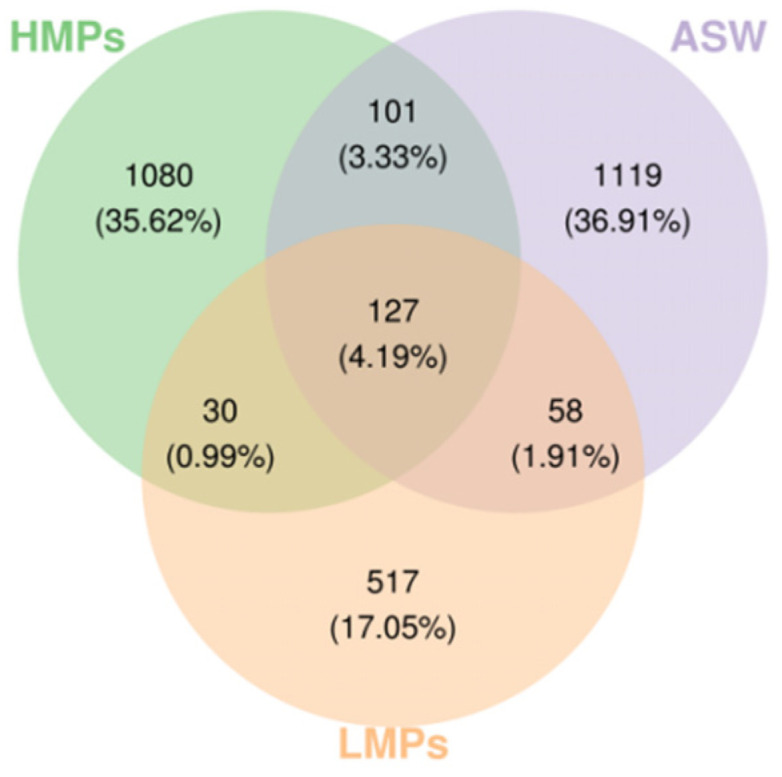
Venn diagram of ASV/OTUs across samples (groups). Each ellipse represents a sample or group, with overlapping regions indicating shared ASV/OTUs between the groups. The numbers within each section denote the number of ASV/OTUs specific to that segment.

**Figure 6 microorganisms-13-00882-f006:**
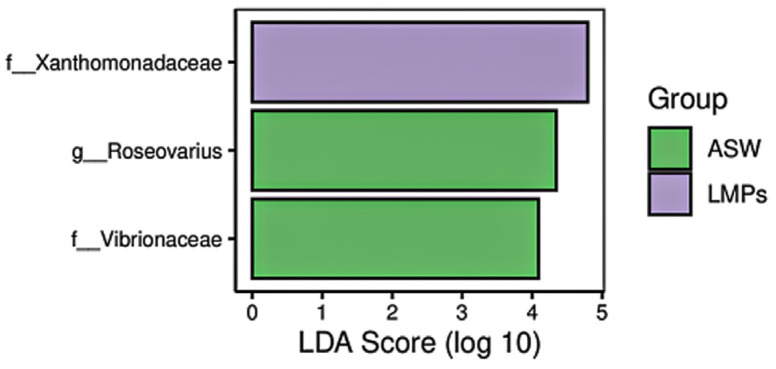
Histogram of LDA effect values for the marker species. The vertical axis lists taxa with significant differences between groups, and the horizontal axis shows a bar chart representing the logarithmic LDA score for each taxon. Taxa are ranked by score, indicating their specificity within sample groupings. The longer the bar, the more significant the difference, with the bar color indicating the group where the taxon is most abundant.

**Figure 7 microorganisms-13-00882-f007:**
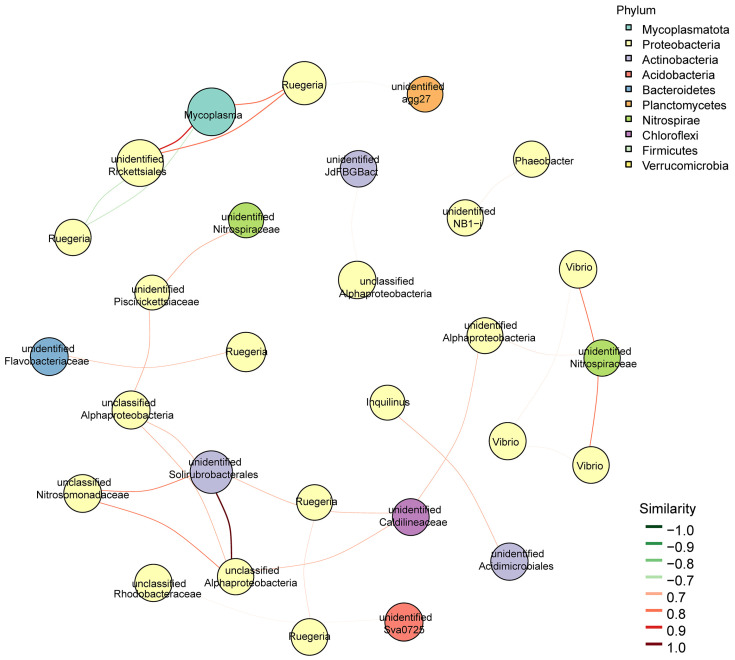
Modular association network diagram. The nodes represent ASVs or OTUs within the samples, with node size proportional to their abundance (in log_2_(CPM/n)). Different colors are used to denote the top 10 modules with the highest number of nodes. Edges between nodes indicate the correlation between connected nodes, with red lines representing positive correlations and green lines indicating negative correlations.

**Figure 8 microorganisms-13-00882-f008:**
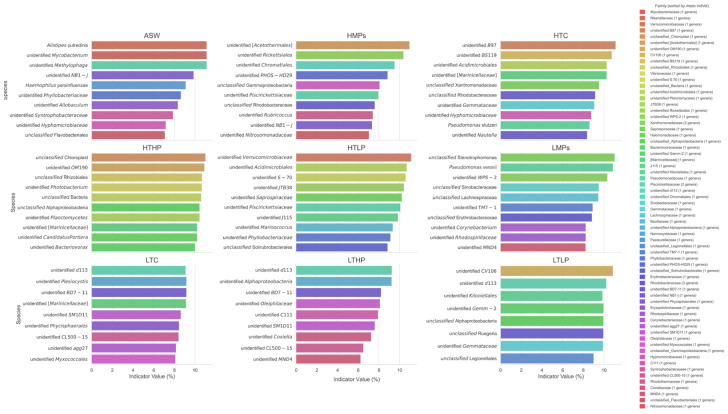
Top 10 indicator species by group. Indicator Value = (Relative Abundance × Specificity) × 100%. The horizontal axis is indicator value (%). Indicator value calculation: 1. Relative abundance: The mean relative abundance of the species in the target group. 2. Specificity: The frequency of occurrence of the species in the target group. 3. Indicator Value = (Relative abundance × Specificity) × 100%; Interpretation: High Relative abundance: The species is abundant in the target group; High Specificity: The species is consistently present in the target group; High IndVal: The species is both abundant and specific to the target group.

**Figure 9 microorganisms-13-00882-f009:**
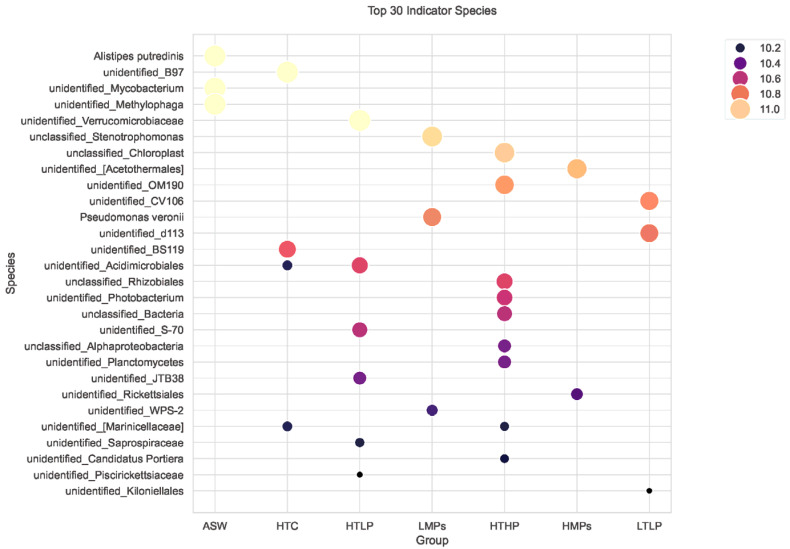
Top 30 indicator species bubble chart. Explanation: X-axis: experimental groups; Y-axis: indicator species; bubble size: indicator value (%) (larger bubbles = stronger specificity and abundance; color intensity: indicator value gradient (darker colors = higher values).

**Table 1 microorganisms-13-00882-t001:** Exposure concentrations of *Aurelia aurita* polyps.

Groups	MP Concentration (mg/L)	TCHCl Concentration (mg/L)
ASW	0	0
LTC	0	0.5
HTC	0	5
LMPs	0.1	0
LTLP	0.1	0.5
HTLP	0.1	5
HMPs	1	0
LTHP	1	0.5
HTHP	1	5

ASW: artificial seawater treatment control group; LTC: low-concentration tetracycline treatment group; HTC: high-concentration tetracycline treatment group; LMPs: low-concentration MP treatment group; LTLP: low-concentration tetracycline combined with low-concentration MP treatment group; HTLP: high-concentration tetracycline combined with low-concentration MP treatment group; HMPs: high-concentration MP treatment group; LTHP: low-concentration tetracycline combined with high-concentration MP treatment group; HTHP: low-concentration tetracycline combined with high-concentration MP treatment group.

## Data Availability

The original contributions presented in this study are included in the article. Further inquiries can be directed to the corresponding author.

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
