# Peer review of "Unraveling the Impact of Microplastic–Tetracycline Composite Pollution on the Moon Jellyfish Aurelia aurita: Insights from Its Microbiome"

_microorganisms, 2025, doi:10.3390/microorganisms13040882_

Round 1
Reviewer 1 Report
Comments and Suggestions for Authors
Unraveling the impact of microplastic-tetracycline composite pollution on moon jellyfish Aurelia aurita: Insights from metagenomics
Overall, the manuscript is limited, although the subject is of good relevance to science. Many sentences should be carefully revised for a better understanding by the reader.
ABSTRACT:
“Microplastics have become a significant contributor to global pollution. Concerns have been raised not only about the ecotoxicity of microplastics themselves but also about their potential to compound antibiotics causing co-harm to organisms and accumulate antibiotic-resistant fragments.”
The first sentence is too short. I suggest improving it by adding more information about microplastic pollution. For example, you could mention the most contaminated sites with plastic pollution. The second sentence is confusing… I suggest rewrite to clarify the meaning.
The introduction of the abstract and the aim don’t have a connection… You should improve it.
“Metagenomic analysis revealed that the environmental concentrations of microplastics and tetracycline did not significantly alter the alpha and beta diversity of the microbiome in A. aurita polyps but affected the relative abundance and composition of certain dominant bacterial species within the community, including Proteobacteria, Actinobacteria, and cell-associated bacteria such as Mycoplasmatota”.
This sentence is too long. Please, separate after A. aurita polyps.
Also, according to Thines et al. 2020, all scientific names must to be write in italic. I suggest to modify this along the manuscript.
Thines, M., Aoki, T., Crous, P.W. et al. Setting scientific names at all taxonomic ranks in italics facilitates their quick recognition in scientific papers. IMA Fungus 11, 25 (2020). https://doi.org/10.1186/s43008-020-00048-6
“Pollutants also affected pathways related to metabolism, cellular processes, and environmental information processing. Furthermore, microplastics may influence life stage traits, which could be crucial for predicting future Aurelia blooms.”
All phrases are disconnected. I really suggest to rewrite the abstract, adding details and linking better the sentences.
INTRODUCTION
This section has a survey of the literature, explaining the main concepts about microplastics, A. aurita species and their microbiome. My comments below can enrich this section.
Line 31-34: Please, add a reference for both sentences.
“MPs are ecologically toxic to both terrestrial and marine ecosystems [1]”.
How toxic? I would like to see examples about the toxicity and potential effects of these pollutants. I noticed you shared some work below, but I’d prefer to see it here in a more concise and direct way.
Line 54-57: please, rewrite this sentence. It is confusing and not clear.
Line 59-61: all sentences are without references.
“Studies utilizing the transcriptomic and microbiological analyses have demonstrated that stimulation by different microorganisms can regulate the A. aurita transcriptome by influencing related gene fragments, including those involved in quorum quenching [13].”
Why this is important for A. aurita? This sentence is missing a complement.
“Although numerous studies have explored the relationship between MPs and the microbiome”.
Please, add some references.
MATERIAL AND METHODS
2.1. Stork Cultures of Polyps
If you can, please add the geographic coordinates, and the collection depth.
23±1℃. Why the temperature was not the same of the maintained (25°C)?
2.3. 16S rRNA Amplicon Sequencing and Analysis
Please, add the PCR conditions and a reference of the primers.
Did you only collect samples at the beginning and end of the experiment? Didn't you have an average exposure time? I missed a topic describing the experiment and the collections carried out in more detail. Please, detail more the topic 2.1.
RESULTS
The phyla names have changed according to Oren and Garrity (2021). It is interesting to actualize the taxa namesof the International Code of Nomenclature of Prokaryotes (ICNP).
Oren A, Garrity GM. Valid publication of the names of forty-two phyla of prokaryotes. Int J Syst Evol Microbiol. 2021 Oct;71(10). doi: 10.1099/ijsem.0.005056. PMID: 34694987.
Figure 1: the colors are not helping to understand the taxa. I suggest modify the colors and the taxa’s legend for genera and phyla.
Line 136: I didn’t understand when you cite Mycoplasma in the figure 2A if the figure is about the classes. Do you mean say Mollicutes? You can increase the font of the graph legend.
Line 188: “Alphaproteobacteria demonstrated a pattern of first declining and then rising.” In my opinion, you cannot affirm this with this figure. Please, reviewed this data.
“Actinobacteria followed a trend of first increased and then decreased”. According to MPs concentration?
You can write more about the results… this is the most important section. Must be complete.
A Species Indicator Analysis (SIA) may be interesting to enrich your results… in this test, you can find the microorganisms that are increased or decreased in treatments with higher or lower concentrations of MPs or tetracycline. I suggest this result for this manuscript. This analysis could be more interesting than Venn diagram and networking correlations.
The histogram can be replaced to supplementary material.
DISCUSSION
This section possessed a survey of related manuscripts and correlations with the results. However, needs to be needs to be reviewed, improved, being more detailed and with more discussions relevant to the work.
4.1. Changes in endophytic flora
This is the first time the term 'endophytic flora' is mentioned. I suggest introducing and explaining this term in earlier sections. Or you can modify the therm.
Line 305: “consistent with previous studies”. Do you have more references about bacterial community?
Line 307-315: all sentences are not connected. I suggest rewriting this part of the discussion, connecting more the sentences and conclusions obtained with the microbiome data, in relation to the different concentrations of pollutants.
Line 333-335: “Previous research on the microbial community of moon jellyfish polyps indicated no significant change in alpha diversity due to ocean acidification [19], whereas community diversity shifted significantly following exposure to radiation [18].” The second work (reference 18) says a significant difference of the microbial community. So, the next affirmation cannot be done. You can modify the text, but exactly as it is, it is not correct.
4.2. Alterations in Metabolic Pathways
Line 376: A. aurita can be fed Artemia sp. nauplii. Did you feed your jellyfish with Artemia? If so, it is important to include this in the methods, including the amount per day and the feeding time.
Do you have any references about the alteration of metabolism in jellyfishes? It might be interesting to mention this, even in other species, in the face of pollution or environmental stress. This topic has not been discussed much with other works, but if there is not, please cite it.
4.3. Potential Life Stage Changing Factor
Good survey and discussion. Maybe, you can perform a link with the topic above, correlating pollutants, and genetic and metabolic changes.
4.4. Ecological Impact
Line 424: an end point is missing.
This topic has a good discussion, but I missed other references on Vibrio, which is a widely studied group of microorganisms, including other marine animals. I suggest doing another bibliographic survey and detailing more about the potential effects of MPs and tetracycline on this microbial group, and also on other marine organisms. Thus, new discussions can be held to better understand the results obtained in this work.
CONCLUSIONS
The conclusions were very similar to the abstract. I suggest rewriting them in a more didactic and complete way for a better closing of the manuscript and understanding by the reader.
Comments on the Quality of English LanguageThe English and many sentences in the text must be rewritten for better understanding by the reader.
Author Response
Comments 1: “Microplastics have become a significant contributor to global pollution. Concerns have been raised not only about the ecotoxicity of microplastics themselves but also about their potential to compound antibiotics causing co-harm to organisms and accumulate antibiotic-resistant fragments.” The first sentence is too short. I suggest improving it by adding more information about microplastic pollution. For example, you could mention the most contaminated sites with plastic pollution. The second sentence is confusing… I suggest rewrite to clarify the meaning.
|
Response 1: I have also rewritten the abstract o address your question about insufficient information. Make its logic clearer and clarify the possible compound hazards caused by the interaction between microplastics and antibiotics.[Line 16-35]
|
Comments 2: The introduction of the abstract and the aim don’t have a connection… You should improve it. |
Response 2: I have revised the abstract as suggested, strengthening the logical flow between key points while addressing both microplastic pollution and its compounded effects with antibiotics. |
Comments 2:“Metagenomic analysis revealed that the environmental concentrations of microplastics and tetracycline did not significantly alter the alpha and beta diversity of the microbiome in A. aurita polyps but affected the relative abundance and composition of certain dominant bacterial species within the community, including Proteobacteria, Actinobacteria, and cell-associated bacteria such as Mycoplasmatota”.
This sentence is too long. Please, separate after A. aurita polyps. |
Response 2: I have revised as suggested. |
Comments 3: Also, according to Thines et al. 2020, all scientific names must to be write in italic. I suggest to modify this along the manuscript.
Thines, M., Aoki, T., Crous, P.W. et al. Setting scientific names at all taxonomic ranks in italics facilitates their quick recognition in scientific papers. IMA Fungus 11, 25 (2020). https://doi.org/10.1186/s43008-020-00048-6 |
Response 3:I have revised as suggested. All scientific names have been italicized as required. |
Comments 4: “Pollutants also affected pathways related to metabolism, cellular processes, and environmental information processing. Furthermore, microplastics may influence life stage traits, which could be crucial for predicting future Aurelia blooms.”
All phrases are disconnected. I really suggest to rewrite the abstract, adding details and linking better the sentences. |
Response 4:I have revised the abstract as suggested, strengthening the logical flow between key points while addressing both microplastic pollution and its compounded effects with antibiotics. |
Comments 5: This section has a survey of the literature, explaining the main concepts about microplastics, A. aurita species and their microbiome. My comments below can enrich this section.
Line 31-34: Please, add a reference for both sentences. |
Response 5:I have revised as suggested.References have been added. [Line 41-42] |
Comments 6: “MPs are ecologically toxic to both terrestrial and marine ecosystems [1]”.
How toxic? I would like to see examples about the toxicity and potential effects of these pollutants. I noticed you shared some work below, but I’d prefer to see it here in a more concise and direct way. |
Response 6: The toxicity of microplastics to various ecosystems has been more clearly expressed and written. [Line 44-46] |
Comments 7: Line 54-57: please, rewrite this sentence. It is confusing and not clear. |
Response 7: I have revised as suggested. [Line 44-46] |
Comments 8: Line 59-61: all sentences are without references |
Response 8: I have revised as suggested. References have been added. [Line 78-81] |
Comments 9:“Studies utilizing the transcriptomic and microbiological analyses have demonstrated that stimulation by different microorganisms can regulate the A. aurita transcriptome by influencing related gene fragments, including those involved in quorum quenching [13].” Why this is important for A. aurita? This sentence is missing a complement. |
Response 9:I have revised as suggested with detailed importance on transcriptomic and microbiological analyses. [Line 85-93] |
Comments 10:“Although numerous studies have explored the relationship between MPs and the microbiome”. Please, add some references. |
Response 10:I have added references as suggested.[Line 116] |
Comments 11:2.1. Stork Cultures of Polyps If you can, please add the geographic coordinates, and the collection depth. |
Response 11:Thank you very much for your valuable feedback. Regarding the suggestion to include geographic coordinates and collection depth, we acknowledge the importance of this information. However, due to the constraints of the current dataset and the scope of our study, we were unable to collect these specific details. We will consider including such data in future studies, where possible, to enhance the comprehensiveness of our analysis. We hope this explanation clarifies our position. |
Comments 12:23±1℃. Why the temperature was not the same of the maintained (25°C)? |
Response 12: The jellyfish Aurelia exhibits notable eurythermy. For Bohai Sea populations [1-2], the documented viable range is 10-25°C with partial mortality observed at 27°C, while experimental temperatures were typically maintained at 18-24°C. Our Hong Kong coastal polyps, normally cultured at 25°C due to their warmer native habitat, demonstrated optimal asexual reproduction activity at 23°C. This temperature change was therefore selected to maximize physiological performance while ensuring cross-population comparability.
1. Duan, Yan; Sun, Ming; Dong, Jing; Chai, Yu; Wang, Aiyong; Wang, Xiaolin; Liu, Xiuze; Wang, Bin; Ji, Guang. Effects of temperature on the growth and asexual reproduction of moon jellyfish (Aurelia coerulea) polyps. Acta Ecologica Sinica 2020, 40, 4404-4412, doi:10.5846/stxb201911152439. 2. Yan Tao, Wang ; Song, Sun ; Chao-Lun, Li ; Fang, Zhang Effects of temperature and food on asexual reproduction of the scyphozoan, Aurelia sp.1. Oceanoologia et Limnologia Sinica 2012, 43, 900 - 904.
|
Comments 13:2.3. 16S rRNA Amplicon Sequencing and Analysis Please, add the PCR conditions and a reference of the primers. |
Response 13:I have added PCR conditions and a renference of the primers as suggested..[Line 171-172] |
Comments 14: Did you only collect samples at the beginning and end of the experiment? Didn't you have an average exposure time? I missed a topic describing the experiment and the collections carried out in more detail. Please, detail more the topic 2.1. |
Response 14:All experimental groups underwent continuous exposure for 185 days. The samples were only collected at the end of the experiment to assess the cumulative effects of long-term exposure. The artifical sea water (ASW) group was used as control group in our experiment which was maintained under identical physicochemical conditions (temperature: 23±1°C; salinity: 30 psu; pH 8.8) with weekly renewal to ensure water quality stability. Detailed exposure regimens and sampling procedures have been expanded in topic 2.2 of the revised manuscript.[Line 152-158] |
Comments 15:The phyla names have changed according to Oren and Garrity (2021). It is interesting to actualize the taxa names of the International Code of Nomenclature of Prokaryotes (ICNP). Oren A, Garrity GM. Valid publication of the names of forty-two phyla of prokaryotes. Int J Syst Evol Microbiol. 2021 Oct;71(10). doi: 10.1099/ijsem.0.005056. PMID: 34694987. |
Response 15:I have revised as suggested. |
Comments 16:Figure 1: the colors are not helping to understand the taxa. I suggest modify the colors and the taxa’s legend for genera and phyla. |
Response 16:I have revised as suggested for better understanding.[Figure 1] |
Comments 17:Line 136: I didn’t understand when you cite Mycoplasma in the figure 2A if the figure is about the classes. Do you mean say Mollicutes? You can increase the font of the graph legend. |
Response 17:Thanks for pointing out the error. I have revised as suggested.[Line 218] |
Comments 18:Line 188: “Alphaproteobacteria demonstrated a pattern of first declining and then rising.” In my opinion, you cannot affirm this with this figure. Please, reviewed this data. |
Response 18:The data is verified and the expression is correct.[Line 220] |
Comments 19:“Actinobacteria followed a trend of first increased and then decreased”. According to MPs concentration? |
Response 19:According to MPs concentration. I have revised for better understanding.[Line 223-226] |
Comments 20:You can write more about the results… this is the most important section. Must be complete. |
Response 20:Thank you very much for your suggestion, I have refined the results for better demonstration. |
Comments 21:A Species Indicator Analysis (SIA) may be interesting to enrich your results… in this test, you can find the microorganisms that are increased or decreased in treatments with higher or lower concentrations of MPs or tetracycline. I suggest this result for this manuscript. This analysis could be more interesting than Venn diagram and networking correlations. |
Response 21:I have revised as suggested.[Line 301-328] |
Comments 22:The histogram can be replaced to supplementary material. |
Response 22:Thank you for your suggestion. We prefer to retain the histogram in the main text as it provides essential visual support for key findings. However, we are happy to move it to supplementary material if the editor considers this necessary. |
Comments 23:DISCUSSION This section possessed a survey of related manuscripts and correlations with the results. However, needs to be needs to be reviewed, improved, being more detailed and with more discussions relevant to the work. 4.1. Changes in endophytic flora This is the first time the term 'endophytic flora' is mentioned. I suggest introducing and explaining this term in earlier sections. Or you can modify the therm. |
Response 23:I have revised as suggested.[Line 302] |
Comments 24:Line 305: “consistent with previous studies”. Do you have more references about bacterial community? |
Response 24:There are other articles about the microbiome of Aurelia, but this article is the closest in terms of the relationship between Aurelia and Mycoplasma. I have revised as suggested.[Line 351] |
Comments 25:Line 307-315: all sentences are not connected. I suggest rewriting this part of the discussion, connecting more the sentences and conclusions obtained with the microbiome data, in relation to the different concentrations of pollutants. |
Response 25:I have revised as suggested.[Line 352-366] |
Comments 26:Line 333-335: “Previous research on the microbial community of moon jellyfish polyps indicated no significant change in alpha diversity due to ocean acidification [19], whereas community diversity shifted significantly following exposure to radiation [18].” The second work (reference 18) says a significant difference of the microbial community. So, the next affirmation cannot be done. You can modify the text, but exactly as it is, it is not correct. |
Response 26:I have revised as suggested.[Line 384-386] |
Comments 27:4.2. Alterations in Metabolic Pathways Line 376: A. aurita can be fed Artemia sp. nauplii. Did you feed your jellyfish with Artemia? If so, it is important to include this in the methods, including the amount per day and the feeding time. |
Response 27:I fed the jellyfish with Artemia. I have revised the method part as suggested. [Line 147-150] |
Comments 28:Do you have any references about the alteration of metabolism in jellyfishes? It might be interesting to mention this, even in other species, in the face of pollution or environmental stress. This topic has not been discussed much with other works, but if there is not, please cite it. |
Response 28:Thank you for raising this important point. While few studies have directly examined metabolic alterations in jellyfish under pollution stress. We have added a brief discussion of this potential connection, highlighting the need for future metabolomic studies in this emerging field.[Line 434-439] |
Comments 29:4.3. Potential Life Stage Changing Factor Good survey and discussion. Maybe, you can perform a link with the topic above, correlating pollutants, and genetic and metabolic changes. |
Response 29:I have revised as suggested. |
Comments 30:4.4. Ecological Impact Line 424: an end point is missing. |
Response 30:I have revised as suggested.[Line 482] |
Comments 31:This topic has a good discussion, but I missed other references on Vibrio, which is a widely studied group of microorganisms, including other marine animals. I suggest doing another bibliographic survey and detailing more about the potential effects of MPs and tetracycline on this microbial group, and also on other marine organisms. Thus, new discussions can be held to better understand the results obtained in this work. |
Response 31:I have revised as suggested by adding infos on Vibrio.[Line 487-492] |

Reviewer 2 Report
Comments and Suggestions for Authors
Questions on the manuscript with ID (microorganisms-3512666-peer-review-v1)
Q1. Line 104: The authors should add the physico-chemical characteristics of the water used for rearing polyps of A. aurita.
Q2. Line 106: Add source of artemia used for fish feeding.
Q3. Lines 110-112: Why the authors used these exposure doses? Did you make LC50 testing prior experimentation?
Q4. Line 114: Source of MPs should be provided.
Q5. Line 115: How the authors confirmed the particle size before the exposure?
Q6. Line 119: How the authors confirm the stability of the exposure doses when they carried out the experiment for a continuous 185 days?
Q7. In figures, you should not repeat the abbreviations. You should only add them for the first time and not repeat after their first appearance in the text.
Q8. In Discussion section, you should delete the sub-titles. Moreover, you should also delete the results.
Author Response
Comments 1:Q1. Line 104: The authors should add the physico-chemical characteristics of the water used for rearing polyps of A. aurita. |
Response 1:As suggested, I have added the physico-chemical characteristics of the water in the text. Please see details in topic 2.1.[Line 131] |
Comments 2:Q2. Line 106: Add source of artemia used for fish feeding. |
Response 2:As suggested, I have added the source of Artemia in the Materials and Methods part.[Line 134] |
Comments 3:Q3. Lines 110-112: Why the authors used these exposure doses? Did you make LC50 testing prior experimentation? |
Response 3:These exposure doses were based on environmental concentrations of these pollutants, as detailed in Section 4.4 of the manuscript.[Line 472-478] In the preliminary LC50 tests, we observed that tetracycline's LC50 was significantly higher than environmentally relevant concentrations. Since this would not reflect real-world ecological scenarios, we focused our formal experiments on concentration ranges that are actually found in the environment. |
Comments 4:Q4. Line 114: Source of MPs should be provided. |
Response 4:The MPs used were demonstrated after. They were spherical polystyrene (PS) particles, 80 nm in size (Tianjin Bessler Chromatography Technology Development Center). Demonstrated at 2.2. Experimental Design. [Line 142-144] |
Comments 5:Q5. Line 115: How the authors confirmed the particle size before the exposure? |
Response 5:They were characterized and identified using TEM (Transmission Electron Microscopy). I added in the article at 2.2. Experimental Design. [Line 144-146] |
Comments 6:Q6. Line 119: How the authors confirm the stability of the exposure doses when they carried out the experiment for a continuous 185 days? |
Response 6:The culture water was pre-equilibrated to 23°C before each water change to maintain thermal stability. The salinity is maintained at 30 psu for each preparation of artificial seawater to maintain salinity stability. Water replacement occurred 1-2 hours post-feeding to ensure consistent pollutant concentrations. During standard cultivation, polyps were maintained in darkness to prevent tetracycline photodegradation. |
Comments 7:Q7. In figures, you should not repeat the abbreviations. You should only add them for the first time and not repeat after their first appearance in the text. |
Response 7:I have deleted the repeated abbreviations as suggested. |
Comments 8:Q8. In Discussion section, you should delete the sub-titles. Moreover, you should also delete the results. |
Response 8: Thank you for your suggestions. We have removed redundant results as advised. However, we retained the subtitles to maintain clear logical flow, as they help organize the discussion themes effectively. We hope this balances readability with your recommendations. |

Reviewer 3 Report
Comments and Suggestions for Authors
This is an interesting paper very well thought out and presented...my only concern was that as I understand this work was carried out with the polyps cultured in plastic dishes...may have ahad an effect on the polyps as there is a leakage of plastics over time. please refer to this in the paper..
another comment is that you use compound effect...I think there may be a synergistic effect as well this is not uncommon as these microbes release chemicals that may affect other microbes (quorum sensing etc) from settling as well and this may have been also affected by the mps
Author Response
Comments 1: Comments and Suggestions for Authors This is an interesting paper very well thought out and presented...my only concern was that as I understand this work was carried out with the polyps cultured in plastic dishes...may have ahad an effect on the polyps as there is a leakage of plastics over time. please refer to this in the paper. |
Response 1: Thank you for your valuable comments and suggestions. We appreciate your insightful observation about potential synergistic effects among microbial compounds. Regarding your concern about potential microplastic (MP) contamination from plastic dishes, we would like to clarify the following points based on our research and existing literature. The containers commonly used in laboratories can be roughly divided into two categories, namely plastic products and glass products.Contrary to common assumptions, studies suggest that glassware may introduce higher levels of microplastic contamination compared to plastic consumables. For example, in the referenced study [1], glassware showed significantly higher MP levels (1,356.9 particles/mL) than plastic labware (6.9 particles/mL). When used in the laboratory, plastic containers are gently cleaned before use to avoid strong friction that may cause the release of microplastics. Plastic materials are lighter, more cost-effective, and widely adopted in our laboratory workflows. During experiments, we strictly maintained low-temperature and light-protected conditions to prevent plastic leaching or degradation.
Reference: [1] Jones, Nina R; de Jersey, Alix M; Lavers, Jennifer L; Rodemann, Thomas; Rivers-Auty, Jack. Identifying laboratory sources of microplastic and nanoplastic contamination from the air, water, and consumables. Journal of hazardous materials 2024, 465, 133276, doi:https://doi.org/10.1016/j.jhazmat.2023.133276. |
Comments 2: another comment is that you use compound effect...I think there may be a synergistic effect as well this is not uncommon as these microbes release chemicals that may affect other microbes (quorum sensing etc) from settling as well and this may have been also affected by the mps
|
Response 2:We fully acknowledge the reviewer's insightful comment regarding microbial interactions. While we indeed considered this aspect, our study primarily focused on characterizing how pollutants alter the endogenous microbiome of Aurelia aurita polyps. The current experimental design did not include specific protocols to differentiate between direct pollutant effects and inter-microbial interactions, nor could we isolate certain unculturable/unidentified strains to test for synergistic effects. These technical constraints limited our ability to mechanistically validate microbial crosstalk, as appropriately noted by the reviewer. We have now expanded the discussion to explicitly address this limitation, and outline strain isolation and co-culture approaches as future directions. |

Round 2
Reviewer 2 Report
Comments and Suggestions for Authors
The authors appropriately responded to the comments and questions raised by the reviewer.